# Bibliometric Analysis of Trends in Mulberry and Silkworm Research on the Production of Silk and Its By-Products

**DOI:** 10.3390/insects13070568

**Published:** 2022-06-23

**Authors:** Domenico Giora, Giuditta Marchetti, Silvia Cappellozza, Alberto Assirelli, Alessio Saviane, Luigi Sartori, Francesco Marinello

**Affiliations:** 1Department of Land, Environment, Agriculture and Forestry, University of Padova, Agripolis, Legnaro, 35020 Padova, Italy; giuditta.marchetti@unipd.it (G.M.); luigi.sartori@unipd.it (L.S.); francesco.marinello@unipd.it (F.M.); 2Council for Agricultural Research and Economics, Research Centre for Agriculture and Environment, Sericulture Laboratory, 35143 Padua, Italy; silvia.cappellozza@crea.gov.it (S.C.); alessio.saviane@crea.gov.it (A.S.); 3Council for Agricultural Research and Economics, Research Centre for Engineering and Agro-Food Processing, Monterotondo, 00015 Rome, Italy; alberto.assirelli@crea.gov.it

**Keywords:** silk, silkworm, mulberry, sericulture, bibliometric analysis

## Abstract

**Simple Summary:**

Over the past two decades scientific research on sericulture, the agricultural activity of silk production, generated a great number of outputs in the form of articles reported and classified by one of the most well-known and used database dealing with scientific literature. This occurrence demonstrates an increasing interest in this sector especially starting from 2000s; results presented in relevant papers showed their applicability even in fields apparently not related to silk production as commonly meant, like medicine, cosmetics, and engineering. To understand how sericulture has been transcending its usual boundaries, which are its current “hotspots”, and links with other fields of study, the authors propose a text-mining based analysis of the outputs of scientific research on sericulture and silk; the final goal is to establish “quantitative” indicators for researchers, entrepreneurs, and scholars.

**Abstract:**

Traditionally, sericulture is meant as the agricultural activity of silk production, from mulberry (*Morus* sp.pl.) cultivation to silkworm (*Bombyx mori* L.) rearing. The aim of the present work is to analyze the trends and outputs of scientific research on sericulture-related topics during the last two decades, from 2000 to 2020. In this work the authors propose a text-mining analysis of the titles, abstracts and keywords of scientific articles focused on sericulture and available in the SCOPUS database considering the above-mentioned period of time; from this article collection, the 100 most recurrent terms were extracted and studied in detail. The number of publications per year in sericulture-related topics increased from 87 in 2000 to 363 in 2020 (+317%). The 100 most recurrent terms were then aggregated in clusters. The analysis shows how in the last period scientific research, besides the traditional themes of sericulture, also focused on alternative products obtainable from the sericultural practice, as fruits of mulberry trees (increment of +134% of the occurrences in the last five years) and chemical compounds as antioxidants (+233% of occurrences), phenolics (+330% of occurrences) and flavonoids (+274% of occurrences). From these considerations, the authors can state how sericulture is an active and multidisciplinary research field.

## 1. Introduction

Sericulture is the agricultural activity that traditionally consists in the cultivation of mulberry trees (*Morus* sp.pl.) to yield leaves that are used for feeding silkworms (*Bombyx mori* L.), reared for silk production. In an ancient period of Human history, the trade of silk represented an important activity that allowed a first kind of globalization, connecting, since about 2000 years ago, Europe to Asia through the so-called “Silk Road” [1,2,3,4]. Nowadays, silk production represents about the 0.2% in value of total world textile production and it is spread over about 60 countries [5]. The most important countries for silk production are China, India, Uzbekistan, Vietnam, Thailand, and Brazil [5].

The term “silk” refers to a class of natural structural proteins produced and spun as long fibrous filament by different Arthropoda, for example Lepidoptera and spiders [6,7,8,9]. The most famous and studied type of silk is produced by the domestic silkworm *B. mori* L. [6]. 

From the beginning of silk manufacturing, silk has been used essentially for textile and medical applications, since efforts in medical research demonstrated how silk could be successfully applied to surgery due to its biocompatibility [10,11,12]. Over the last few years, there has been a growing interest in the engineering of biomaterials capable of mimicking the structure and characteristics of tissues and several studies have focused on the use of natural silk fibroin [10,13,14,15], due to its high biocompatibility when separated from sericin [13]. In recent years even sericin acquired a high commercial value [16] thanks to a series of properties that make it suitable for application in the pharmaceutical, cosmetic and food sector [17]. In fact, sericin has been shown to have several biological activities, such as antioxidant and anti-tyrosinase properties [17,18,19,20] and it is used for controlled drug-releasing biomaterials to promote stability and prolonged administration of drugs, enzymes and insulin [20,21,22].

To produce commercial silkworm eggs male and female larvae of pure strains are reared in a dedicated environment to create the parental lines that are then crossed among them [23]; the most common parental lines belong to Chinese and Japanese strains [24,25]. Hybrids and polyhybrids are in general more productive than pure strains and more resistant to biotic and abiotic stress; Nevertheless the management of environmental parameters, as air temperature and relative humidity, is fundamental for the optimal development of reared larvae since these factors heavily affects larval metabolism [26].

Traditionally, *B. mori* larvae feed on fresh leaves of the white mulberry (*Morus alba* L.) [27,28], although even leaves of other species belonging to *Morus* genus can be used [29]. Leaves of *M. alba* contain the optimal amount of nutrients and compounds that are fundamental for the growth and development of *B. mori* larvae and for their silk cocoon spinning. Besides being the only feeding source for silkworms, several studies have also been focused on alternative uses of by-products from *Morus*: leaves, roots, stems, and fruits of different species belonging to the *Morus* genus (in particular, *M. alba* and *nigra* L.), contain relatively high amounts of phenols [30,31,32], in particular flavonoids [32,33,34] with predominant quercetin and kaempferol glycosides as well as anthocyanins [35]. Different experiments showed how mulberry cultivation can represent an interesting and sustainable source of these compounds for nutraceuticals, which are usually contained in the mulberry leaf powder and mulberry fruit juice [36,37].

The previous paragraphs offer a brief overview about the whole compart of the silk production and show how the theme is wide and linked to several other topics, as engineering, medicine, and nutrition. In this complex framework, a text-mining analysis as proposed by Cogato et al. [38] and Ferrari et al. [39] seems to be appropriate to appreciate the interconnection among the hotspots of the research over the past two decades, from 2000 to 2020. This kind of analysis, in fact, allows to objectively identify and weigh the most important topics in a specific research field and to study how they interact.

The aim of the present analysis is to provide a general and comprehensive review of the state-of-the-art of literature about sericulture in the fields of agriculture and biological sciences. The specific objectives of this work were to: (i) provide a description of the temporal trend of publications over the past 20 years; (ii) identify the most important topics to which the research for sericulture field has been mainly directed; and (iii) analyze the most important links among topics; this aspect is very important since sericulture is based on the interactions of different production chains and processes. In order to obtain such results authors decided to perform a quantitative analysis that is the most effective and objective methodology to achieve the above-mentioned specific goals.

## 2. Materials and Methods

As first step for the bibliometric analysis, the authors downloaded a collection of documents indexed by the SCOPUS database, taking advantage of the *advanced search* to precisely define the topics. The analysis of scientific documents contained in the SCOPUS database allowed the authors to understand how the interests of researchers developed and changed in the past years.

The authors performed a *text mining* process of the selected documental collection, in order to derive significant numeric indices and information, by analyzing unstructured (textual) information. The statistical analysis of these indices provides a key to the text interpretation, obtaining high-quality information useful for content interpretation. The words appearing in the title, keywords, and abstract of all articles were first pre-processed in the Microsoft^®^ pre-installed Notepad environment and used for the creation of basic graphs in Microsoft^®^ Excel^®^; the more elaborated graphical representations were obtained using Gephi (Gephi^®^ Consortium, Compiegne, France), an open-source software developed to perform network analysis.

### 2.1. Article Selection

The analysis was based on the words “*silkworm*” and “*mulberry*” both as parts of the process that leads to silk production. Moreover, to include articles containing the derived forms of those two nouns, the scripts “silkworm * ” and “mulberry * ” were used for the initial research in SCOPUS, since the asterisk “*” indicate any word-related declination: e.g., “mulberry”, “mulberries”, “mulberroside” and so on (Table 1). With the initial examination, the program selected the articles that contain the string “silkworm” and “mulberr” and their derived terms in the title, keywords, or abstract. The authors applied some filters (in Table 1, Filter application n.1) for a more pertinent selection of the articles: the field was limited to “Agricultural and Biological Sciences”, the language was limited to “English” and the type of document was limited to “Scientific Article”; the time span considered was 2000–2020. According to Author’s expectations, this analysis resulted in finding many articles (more than five thousand) even with the application of the described filters.

The authors then applied a second series of filters to reduce misleading results. In particular, the word “*Antheraea*” was excluded from the research since it is linked to the topic of silk production but referred to a wild Lepidopteran rather than to *B. mori*. Articles collected by popular and dissemination journals were also excluded by the research. The new script is shown in Table 1 at Filter application n.2 row. With the applications of those new filters, the number of articles dropped to about 4600.

The download procedure from the SCOPUS database was repeated for each year between 2000 and 2020; the .csv format was chosen for the downloads.

### 2.2. Article Elaboration

The results of the research were converted and saved as .txt files to have all the data stored in a file format that could be easily processed. The first step of the data pre-processing was the so-called *tokenization*, a procedure through which the sentences are divided into their essential components (i.e., single words). All the other parts of the text as punctuation marks, hyphens, brackets, and others special characters were removed. The authors run a further elaboration to convert all capital letters into lowercase and to identify and convert singular/plural nouns (e.g., mulberry/mulberries or leaves/leaf) and synonyms (e.g., white mulberry and *M. alba*).

All .txt files obtained at the end of the previously described process were imported into Microsoft^®^ Excel^®^ (Microsoft 365 MSO Version 2111 Build 16.0.14701.20254 64-bit software, Microsoft Corporation, Redmond, WA, USA) to order all the single selected words and to count how many times each one appeared. This elaboration was necessary for the identification of the more frequent words in each considered year of the past two decades. Using Excel, the 100 most relevant words were identified and used for the subsequent analysis. Another interesting function provided by the Excel^®^ software is the so-called *sparkline graphs* that creates a small graph into a single cell; the authors used this function to create time series graphs for all the analyzed nouns, thus recognizing in a fast way which words increased in importance during the analyzed time. As final step, the results were imported as .csv dedicated files in the Gephi software [40], which is a free open-source software that allows for the creation of complex graphical representation of word associations. To create a connection graph in Gephi, the user has to import into the software a .csv file containing information about the *nodes* (here, the most frequent nouns derived from the previous elaboration) and another .csv file containing information about the *edges*, the connection thanks to which each node is related to the others. Gephi considers the edges as vectors, directed or undirected, each one with its specific *weight*, which is generated as described in Section 2.2.1.

#### 2.2.1. Combination Matrix

A word–word connection matrix was built as detailed by Ferrari et al. [39], based on the list of the 100 most recurrent words. The matrix had 100^2^ cells and, that was the result of a total of 10,000 couples. Starting from this matrix, the authors built a dedicated connection matrix in which, for each couple of words (e.g., *mulberry-silkworm*) the number of articles that contained both the nouns was indicated; the connections of words are not directional, namely the order of the words forming the couple was not considered (e.g., *mulberry-silkworm* is equivalent to *silkworm-mulberry*) in order to avoid duplications. As a result of the matrix, 4950 couples of nouns were obtained.

#### 2.2.2. Cluster Definition

Besides the analysis for the combination matrix and therefore for studying how the nouns are interconnected, the authors performed a so-called *cluster analysis*. The aim of the cluster analysis was to define a rule (or a feature, or a set of features) that allows to create groups containing a set of objects (the words in this case) and sharing one or more previously defined features: in this precise kind of cluster analysis the aim is grouping words according to the thematic area they belong to. When the main topic can be subdivided in several interrelated parts, as the one here-considered, cluster analysis allows to study and characterize the relationship among the different sections of the main topic: for example, how the cultivation of mulberry trees connects to silkworm rearing and to silkworm applications. As this analysis is only qualitative, by further differentiating results for the years or for the geographical area, it is possible to describe in a precise way the evolution of the relationship among clusters. It is worth noting that there are no fixed rules to define clusters, but the experience of the authors and the definition of precise rules, useful to minimize equivocations, allow the authors to discriminate the clusters. In the present work, the authors propose the different activities of the sericultural chain as criteria to define the clusters and, in this way, five clusters were identified: Cultivation, Silkworm, Rearing, Process and Product. All the words regarding the mulberry cultivation group in the cluster “cultivation”: for example, words pertaining to the agronomic techniques and to the plant biology. The cluster “silkworm” collects all the words referring to the biology of silkworms (e.g., strains or different stages of their life). In the cluster named “rearing” the authors pooled all the words regarding the activity of silkworm rearing, including diseases or environmental factors (e.g., air temperature) that are fundamental for the insect development. The cluster “process” includes all terms that are linked to the industrial processing of products deriving from both silkworm rearing and mulberry cultivation. The cluster “product” groups all the words that describe the production obtainable by both mulberry (e.g., important metabolites) and silkworm. Although these clusters and rules were defined to avoid ambiguities and overlapping, the authors used the combination matrix and the specific research of documents on SCOPUS to define which words best fit to different cluster. Table 2 shows the cluster composition.

The authors classified the most frequent words by assigning to them a score calculated as weighed average of the word in the considered years. The authors calculated, per word and per year, the ratio of the number of occurrences to the total number of publications in the sericultural field; in this way the authors could objectively characterize the relative importance of the selected word in a certain year and study its trend over the considered interval of time. The authors assigned the highest weight to the most recent years to pay more attention to the last trend of research in sericulture. The score for each word was calculated as proposed by Ferrari [41] (see Equation (1)).
(1)$$W_{i}=\frac{\sum_{i=1}^{21}w_{i}\cdot \frac{o_{i}}{S_{i}}}{\sum_{i=1}^{21}w_{i}},$$

In Equation (1), *w_i_* is the weight of the *i*-th year (from 2000 to 2020), *o_i_* is the absolute number of occurrences in the *i*th year and *S_i_* is the total number of publications in the *i*th year.

The authors analyzed the impact of the five most frequent terms of each cluster, according to Equation (1). To this end, the impact of the papers related to such terms was quantified in terms of Hirsch’s h-index. Thus, for a given term, the h-index has been calculated here by counting the number *h* of publications including that term and cited at least that same number *h* of times. The average was then computed for the five terms of each cluster.

The conceptual flux of the analysis is represented in Figure 1.

## 3. Results

### 3.1. Analysis of the Trends

As first analysis, the authors considered the number of articles published per year in the cumulative research on mulberry and silkworm and the ratio to the total number of publications in the agricultural and biological science field as an indicator of the general interest for the topic. As can be seen in Figure 2, the number of publications recorded a constant and quite rapid increase between 2000 and 2011, with an average increase of 20 articles per years. Between 2012 and 2015 a drop in the number of publications per year can be detected, reaching a relative minimum in 2015. Then, the number of publications per year increased until 2020.

On the basis of Figure 2, in the time span from 2000 to 2020 two main peaks of interest for the silkworm and mulberry topics can be detected, in 2005 and 2011, also considering the ratio of the number of publications regarding the mulberry and silkworm topics to the total number of publications in the agricultural and biological science field. From 2012 to 2018, the ratio of the selected topics to the total number of publications decreased, reaching its minimum in 2015, and then remained stable for almost five years, although the absolute number of publications in the silkworm and mulberry topics increased.

To characterize the geographic distribution of the research in the sericultural topic, an analogue analysis was performed considering the number of publications per country. Figure 3 shows the top five countries for number of publications in the 2000–2020 period.

The three most important countries for the research in the sericultural topic are China, where the number of publications increased from 3 to 180 per years, India and Japan, where the interest increased more or less until 2010 and then decreased.

A more specific analysis about the geographical extent of the research is the one focused on collaborations among Universities and Research Institutes from different countries that aims to determine how the chosen topic can generate collaborations in the scientific community. Figure 4 shows in a graphic way how the web of collaborations for the sericultural topic is well distributed in the World among different countries. More detailed information about collaborations among different countries are presented in Table 3, where data about the first five countries for the total number of collaborations that led to articles publication between 2000 and 2020 are summarized; data of collaborations were derived from the affiliation metadata of the SCOPUS database.

As it can be noticed, the first country for number of collaborations is Japan with a total of 142 articles derived from cooperation. Japan collaborated with 37 different countries, with an average of 3.9 articles derived from the collaboration with each one. The USA are the first country for number of partners of research, but with a lower total amount of published articles and average number of collaborations per country.

### 3.2. Analysis of the Most Important Words

The authors then analyzed the collection of the most recurrent words previously derived as described in Section 2.2. The authors started the analysis on the list of the most recurrent words both in a cumulative way and considering their evolution over time too; this study was performed to characterize in a better way the research outcomes during the last two decades.

The authors choose three different parameters to characterize the evolution over time of the selected 100 words objectively. For the computation of those parameters, the authors used Microsoft Excel. The first parameter is the *mean slope* of the regression line that fits the number of publications per year; this parameter is calculated by Microsoft Excel dividing the variation on *y*-axis by the variation on *x*-axis. This parameter allows the authors to understand the rate of increment or decrement of the selected word during the investigated span of time. The second parameter is the *Pearson correlation coefficient*. This parameter is adimensional and ranges between +1 and −1. It expresses the linearity of the trend (i.e., evaluates whether the increment or decrement of the occurrences, for the selected word, are constant over the years). The third parameter is the last *five years relative change*, calculated as ratio between the average number of occurrences per word from 2016 to 2020 and the average number of occurrences from 2000 to 2015. Table 4 resumes some of the most cited nouns and their associated parameters.

Then, the same parameters used for characterizing words in the previous analysis were calculated out of the normalized number of occurrences per year; the normalized number of occurrences was calculated as ratio of the absolute value of the occurrences of the selected word and the absolute number of publications in the corresponding year. Results for the same nouns of Table 4 are reported in Table 5.

### 3.3. Cluster Analysis

The 100 top frequent words have been grouped into six clusters. With reference to Equation (1), the authors used the weight assigned to each term to define the relative weight of the cluster to which that specific word belongs. Therefore, the relative weight of each cluster was calculated on the basis of the total sum of the relative weights of all the *n* words contained in the cluster itself. Table 6 shows the constitution of each cluster, with the relative weights of all the considered words.

With reference to Table 4, the most important clusters are, in descending order, “silkworm”, “cultivation”, and “product”. The “silkworm” cluster represents about one half (46.4%) of the total occurrences considered for this analysis. The authors grouped in this cluster all the nouns that refer to the biology of the silkworm *B. mori*. In this cluster, about one forth (26.0%) of occurrences are in general about the silkworm itself, respectively with the words “silkworm” (15.3%) and “*B. mori*” (10.7%). In absolute terms the occurrences of the word “gene” (12.6%) saw a rapid increase (slope = 18.2) that was constant and quite linear during the considered period (Pearson coefficient = 0.85); this trend is maintained considering the ratio of occurrences over the number of publications per year in sericulture. In relation to “gene”, the word “expression” (7.4%) shows a significant increasing both in absolute (slope = 14.3) and relative terms (slope = 3.45) Another important theme in the biology of silkworm is represented by the sex identification, and in this analysis this topic represents the 3.3% of the weight of the cluster, respectively with words “female” (1.5%), “male” (0.9%), and “sex” (0.8%). Other important nouns belonging to this cluster are “larva”, that represents 5.8% of the total weight of the cluster, “cocoon”, with 3.4%, “shell”, with 0.9% and “spinning”, with 2.9%. The cluster named “cultivation” groups all the aspects linked to the mulberry cultivation and mulberry field management; its overall relative weight is 23.6%. The most recurrent word is “mulberry”, with about one third of weight of the cluster (32.2%). In particular, the most present one is the white mulberry, *M. alba* (L.), whose weight in the cluster is 6.5%; then the second most studied species is the black mulberry, *M. nigra* (L.), with 2.9% of the total weight of the cluster. For both the species, the increment of interest in the last two decades in absolute terms showed a constant and quite linear increase, with a slope = 6.4 for *M. alba* and 3.3 for *M. nigra* and a Pearson coefficient = 0.92 and 0.91 respectively. *M. nigra* also shows the highest increase of interest in the last five years, trend confirmed both in absolute terms (+161% vs. +118%) and in relative terms (+94% vs. +50%). Other two important words of this cluster are “leaf” (15.2%) and “fruit” (6.5%). The noun “leaf” shows a good growth but not so constant in time in absolute terms (slope = 10.2 and Pearson coefficient = 0.73). However, in relation to the increasing number of publications in the sericulture field the interest for it decreased (slope and Pearson coefficient near to zero but negative and relative increment in the last five years ±0%). At the opposite, the interest of researchers for mulberry fruit increased both in absolute terms (slope = 7.3, Pearson coefficient = 0.92 and relative increment in the past five years = +134%) and in relative terms (slope = 1.99, Pearson coefficient = 0.85 and relative increment in the past five years = +77%). The third most relevant cluster is “Product”, which is dedicated to the products obtainable from both mulberry cultivation and silkworm rearing. The relative weight of this cluster is 22.2%. In this cluster, the most important word is “protein”. The importance and interest for the word “protein” arose in a rapid and linear way during the last two decades (slope = 12.6 and Pearson coefficient = 0.90) in absolute terms; in relative terms this increase is not so high, with a peak of interest during the beginning of 2000s (see graph in Table 5). An analogue consideration could be made for the word “silk” that shows a quite high interest in absolute terms, but this increase in interest is biased by the increasing number of publications per year in the sericultural field. In fact, when we perform the normalization of the trend on the number of publications per year, we can see that this interest has been constant for researchers. Linked to the word “silk” there are nouns referring to its principal components, “fibroin” (1.5%) and “sericin” (1.5%). In this cluster, another important category of words is represented by beneficial compounds that could be extracted from mulberry leaves. These words are “antioxidant” (6.9%), “anthocyanin” (3.4%), “phenol” (3.5%), “flavonoid” (2.3%) and “polyphenol” (1.4%) and their sum weighs about 17% of the whole cluster. In particular, the interest for this group of words rapidly increased in the past five years, both in absolute and relative terms, as attested by the parameter of relative increase reported in Table 4 and Table 5.

As last analysis, the authors evaluated the temporal trend of the aggregated number of occurrences per cluster. Furthermore, the trend of the normalized total occurrences as a ratio to the number of publications was analyzed. The results are reported in Table 7 and Table 8.

From the analysis of Table 5 and Table 6 some interesting considerations can be derived. In particular, the “silkworm” cluster, in absolute terms, has the highest value of slope, followed by “product” and “cultivation”; the cluster “process” is the one with the major relative increase of publications in the last five years. On the other hand, when we normalize the number of publications per year, the silkworm cluster evidences a negative slope, which, although linked to the relative increment of the last five years, testifies a slow decreasing interest in this topic. “Cultivation”, “process” and “product” are the cluster that maintain a high relative interest in the field of sericultural research.

### 3.4. Impact Factor Analysis

The objective of this analysis was to derive sound information about the impact that the most frequent terms generated on scientific research. The proposed impact factors of considered word are reported in Table 9.

As reported in Table 9, the “cultivation” and “product” clusters are the ones exhibiting the highest impact, highlighting a vivid interest by the scientific community on these topics. Also, the “silkworm” cluster gives evidence of a significant impact: in this case, the high h-index should also be related to the highest number of papers published on this topic since 2000. A lower interest is apparently arising from research related to the process or to entomological aspects. On the other hand, the specific terms, “acid” (97) and “gene” (95) gained the highest impact, having collected more than 35,000 citing documents each.

### 3.5. Interrelationships among Words

The objective of this analysis was to derive sound information about the interrelationships among the analyzed words. To achieve this aim, each of the 100 most frequent words previously mentioned was coupled with each of the remaining 99 words, thus generating a table of 4950 possible combinations (avoiding repetitions). Each noun of this collection of paired words was studied in terms of total number of occurrences for the last considered 20 years.

As reported in Table 10, “silkworm” is the cluster with the highest number of co-occurrences, both inside the cluster itself and with other clusters; the maximum number of occurrences was reached by the interaction of “silkworm” with itself (61,864), with the cluster “cultivation”, (20,626) and with the cluster “product” (8112), attesting the great attention of the scientific community to themes focused on the biology and life cycle of silkworms and to the production that could be derived from them. The cluster “cultivation”, as expected, is the second one for number of co-occurrences; as for the cluster “silkworm”, the co-interactions are mainly concentrated on the cluster “cultivation” itself and on the cluster “product”. In this case, the interaction with the cluster “process” is also high.

Then, the authors proposed a brief analysis of the most recurrent couples of nouns, independently on the original cluster, to stress the importance of the most relevant topics. The couple of words with the highest number of co-occurrences is silk-silkworm (2435) *Bombyx*-*B. mori* (2097), and *B. mori*-Silk (1953). The first couple for co-occurrences related to the theme of mulberry cultivation is the couple leaf-mulberry (924) followed by *Morus*-*M. alba* (666); several couples of terms regard the area of genetic studies, as Silk-Gene (1366), Silkworm-Gene (1297), *Bombyx*-Gene (1245) and *B. mori*-Gene (1243).

Taking advantages of the Open-Source software Gephi, the authors generated the map of the interrelationship between the 100 most frequent terms showed in Figure 5.

## 4. Discussion

The main core of this research is represented by a text-mining analysis focused on the words in the title, abstract, and keywords of articles related to silk production. Different considerations can be drawn by the analysis of the most frequent nouns, which were previously identified and evaluated. Taking advantage of a procedure algorithm already tested by Cogato et al. and Ferrari et al. [38,39], the most significant relationships among nouns were recognized. Additionally, by including time parametrization, it was also possible to characterize the evolution of the research interests in sericulture. Thanks to the large number of documents considered in this study, a statistically robust analysis of research was achieved, supporting hypotheses on influences and trends in sericulture

From the characterization of the number of released publications in sericulture (see Figure 2), it is worth noting that the absolute number of articles published per year has been increasing since 2000; such trend reflects the great interest in the topic and the interdisciplinarity of the matter. By considering this trend, two main periods can be identified: from 2000 to 2011 the trend was constantly increasing, both in absolute number and in relative percentage (considering the total number of publications indexed by the SCOPUS database), reaching a peak in 2011; the second period began in 2012 and exhibited a decrease followed by an increase until 2017. Currently, the interest is still apparently growing in absolute terms. In relative terms, the interest appears quite stable; this is due to a rather great relative increase in the total number of publications per year (from about 62 k in 2000 to more than 200 k in 2020, documents indexed by the SCOPUS database). The next decades should be monitored to determine whether these new trends are stable, or they are not. From a geopolitical and temporal point of view, such a trend should be explained by two main reasons: the decreasing number of publications per year from India, and the increasing number of publications per year from China, which is now the leading country for the research in the sericultural field.

From the analysis of the trends of single words and clusters, interesting and more specific information have been extracted. From an aggregate point of view, considering the clusters, the main arguments of investigation of the research in the past twenty years were the study of the *Morus* cultivation, industrial processes, and products that could derive from sericulture, from both *Morus* cultivation and *B. mori* rearing.

From the analysis of the impact of the most important words of each cluster, it is interesting to note how those words are able to generate a considerable impact in the frame of sericulture, with an overall average impact factor of about 67.

The cluster “silkworm”, related to the silkworm biology, is the largest one in terms of number of included words (according to the data of Table 6, 46.4%) but during the last decade, and in particular in the last five years, the interest of the scientific research decreased relatively to this sericultural main topic (Table 8). In particular, the most important words of the cluster are “silkworm” (15.3%) and “*B. mori*” (10.7%), also characterized by considerable high values of the impact factor (respectively 78 and 77); this is easily understandable since the work on the living organism is the main focus of the research on this topic. The silkworm has always been an important laboratory tool for genetics and physiology of insects; in particular the independent genome sequencing of 2004 by a Chinese [42] and a Japanese team [43] and the integrated silkworm database of the end of 2009 [44], gave probably place to the peak of publication on sericulture recorded in 2011. However, from 2012 (the year of FAO experts’ consultation in Rome about entomophagy “Insects to feed the world”) new insects started to be studied in relationship with the new trend of edible insects as perspective feed and food for the future; for example, *Tenebrio molitor* genome was sequenced in 2020 [45]. Exploitation of edible insects gave rise to a new impulse to research on the physiology of new and less known insects as *T. molitor* and *Hermetia illucens*, probably resulting in a diminished interest for traditional insects used as laboratory tools like silkworms.

However, silkworm genetics, still accounts for about 22% of the weight of this cluster. The Nucleopolyhedrosis virus (NPV) has also been used for genetic studies, and in particular as a vector for the recombinant protein technology [46,47,48]. NPV research could, therefore, be considered for its importance in determining both methods for biotechnologies and silkworm physiology studies or in establishing correct rearing techniques for disinfection, disease prevention and rearing facility planning. For this reason, NPV research and other studies on silkworm diseases like pebrine (*Nosema bombycis*) or white muscardine (*Beauveria bassiana*) could be regarded as horizontal literature common to the cluster “Silkworm” and “Rearing”.

Another important theme of the silkworm biology is represented by the sex identification. Indeed, in order to obtain a highly-productive polyhybrid, the best male and best female individuals of different strains are reared and bred. As stated by Raj et al. [23], different solutions for non-destructive sex-sorting methods for silkworms have been proposed [23,41,49]. In fact, an efficient sexing method has not been found yet. Three other important terms of the cluster “Silkworm”, counting about 10% of the weight of the cluster, are “larva”, “cocoon” and “shell”, the raw materials for farmers who rear the larval stage of silkworm and obtain, at the end of their fifth larval instar, fresh silk cocoons.

The cluster “cultivation”, related to mulberry cultivation, is the second most important one in terms of the relative weight of the included words (according to the data of Table 6, about one fourth). Since mulberry is the general word for the identification of *Morus* sp. pl. plants, the most interesting and reliable results are about specific species. As stated before, silkworms feed mainly on *M. alba* leaves [27,28], and the term “*Morus alba*” showed a high increases in the past twenty years. The term “*Morus nigra*” recorded a lower increase, located in particular in the 2016–2020 period. The term “leaf” is the second most important word in this cluster, and this is related to the fact that mulberry leaves represent the only source of food for silkworms [27,28]. It is worth noting how “leaf” is a word that has not exhibited an increase, as for others considered terms (Table 5). The explanation of this trend could be related to the fact that *Morus* is currently studied much more for fruits and active ingredients for pharmaceuticals than as feed for the silkworm, for which artificial diets are an innovative and attractive matter of research.

In the “product” cluster, the most recurrent word is “protein”, which represents the most important class of nutrients in mulberry leaves, the only source of nitrogen for silkworms. Thus, silkworms use the proteins obtained through dietary ingestion in order to extract the amino acids necessary to synthesize other required proteins both for their growth and for spinning cocoons [50]. Proteins are, therefore, also important products obtained from cocoons (fibroin and sericin) and from silkworm pupae. Recombinant proteins achieved from engineered larvae also represent a “hot spot” of sericulture [46,47,48,51]. It is worth noticing how the nouns “acid” (97), “extract” (82) and “protein” (82) are characterized by very high values of the impact factor here proposed, attesting a great interest of the scientific research about the topics related to these nouns.

The “process” cluster is the one with the lowest number of occurrences and thus with the lowest weight (1.6%). Nonetheless, an actual interest on the topic can be recognized, especially in the last five years (2016–2020), when the cluster gave evidence of the highest increment. New industrial processes are currently required for recombinant protein purification, for fibroin and sericin dissolution and reconstitution, to build 3D scaffolds, biomedical products, biocompatible implantable devices and biofilms. Furthermore, to purify mulberry organic compounds, fat and proteins from pupae, new methods and processing technologies are necessary and this explains the increase of the last years linked to the emerging fields of sericultural biotechnologies, pharmaceuticals from mulberry, and edible insects [52,53,54].

In the “rearing” cluster the two most relevant themes are those related to the words “feeding” (already discussed above) and “temperature”. The word “temperature” is the one with highest weight in the cluster (13.5%) and this trend is somehow confirmed also by the high attention paid by farmers as well as by researchers to the temperature control for the sericultural practice [26,55,56]; in fact, some authors proposed sensor-based systems that allow the automatic control of temperature (concurrently with other environmental parameters) [57]. For the same reason, a clear interest is also focused on other two strictly related nouns namely “heat” (4.5%) and “humidity” (2.7%). We should also highlight that climatic changes have impacted on all the agricultural activities and sericulture was also much affected by the extreme environmental conditions recorded in the last years.

## 5. Conclusions

Due to the renovated high interest in natural fibers (on the SCOPUS database, from 367 document published in 2000 to 1768 document published in 2020), in this work the authors propose a detailed analysis of the outcomes of the scientific research on silk. The results of the analysis showed a high number of documents related to classical themes of silk production, as the quality of silk, the biology of *B. mori* and the cultivation of *M. alba*.

The analysis has highlighted some gaps of knowledge in sericulture, in particular a low amount of research is related to the themes of automation of processes. The automation applied to time-consuming tasks as feeding or environmental control in rearing rooms could be successfully carried out through new possibilities offered by the application of Machine Learning and Artificial Intelligence [58] and constitute an important research field for the future sericulture.

## Figures and Tables

**Figure 1 insects-13-00568-f001:**
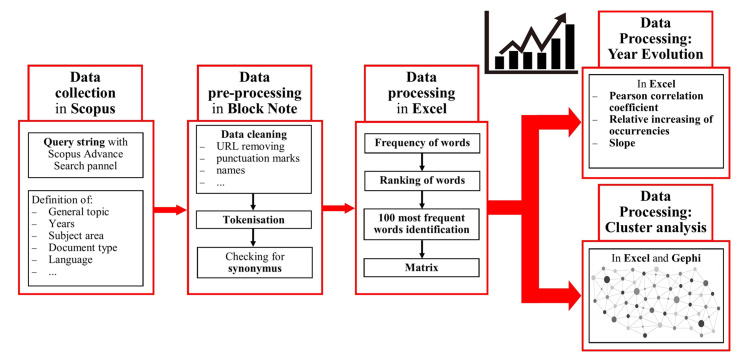
Scheme of the whole analysis process of the text data.

**Figure 2 insects-13-00568-f002:**
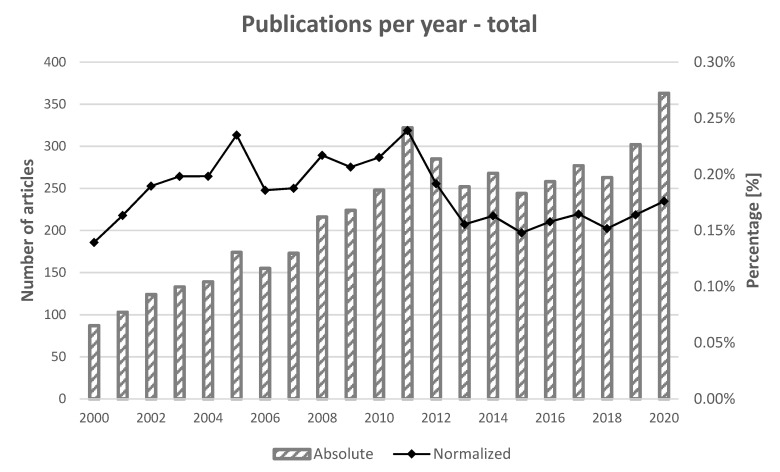
Publications per year (bars) and ratio between publications on the mulberry and silkworm topic in the sector “Agri” and total publications in Agricultural and Biological Science field (line).

**Figure 3 insects-13-00568-f003:**
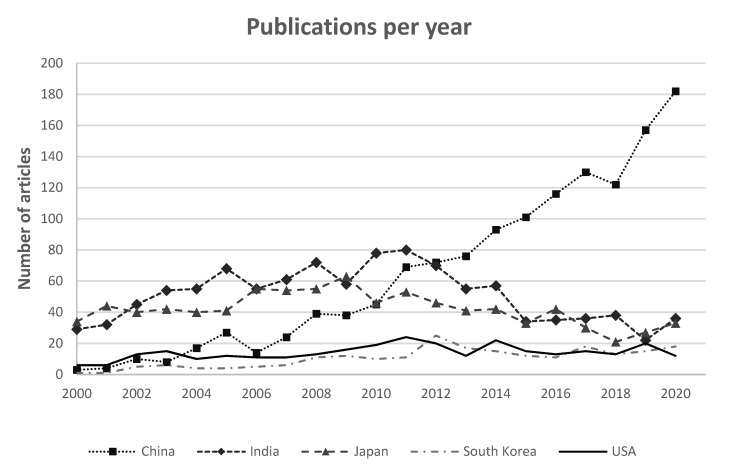
Geographic distribution of interest for the sericulture topic.

**Figure 4 insects-13-00568-f004:**
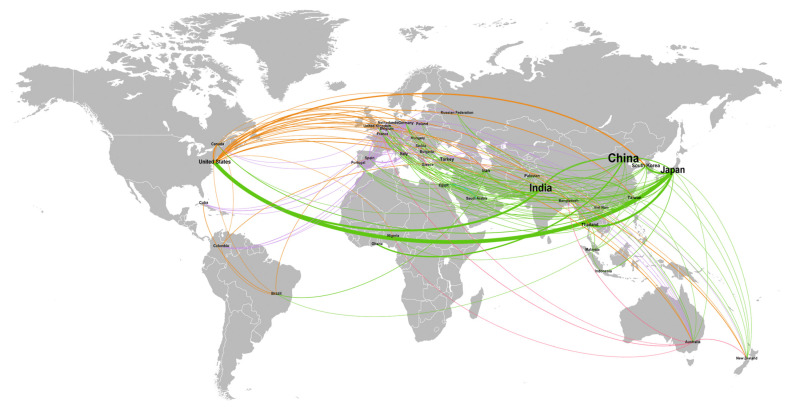
Geographic extent of collaborations for research in sericulture-related topics.

**Figure 5 insects-13-00568-f005:**
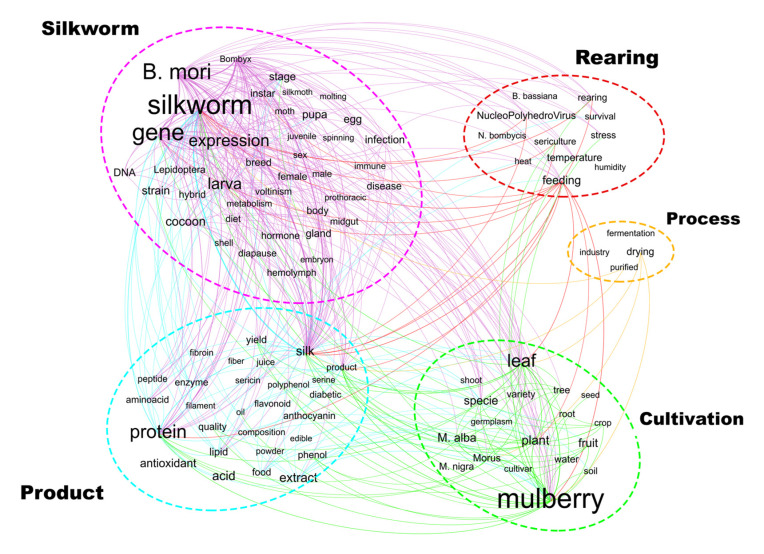
Interrelations among the top 100 most frequent words. In this picture, the curved lines represent relevant interrelations (number of co-occurrences is greater than 100).

**Table 1 insects-13-00568-t001:** Scripts for the extraction of research papers in the SCOPUS database.

Step	Script	Number of Papers ^1^
Initial research	TITLE-ABS-KEY (mulberry *) OR TITLE-ABS-KEY (silkworm *)	23,580
Filter application n.1	TITLE-ABS-KEY (mulberry *) OR TITLE-ABS-KEY (silkworm *) AND (LIMIT-TO (DOCTYPE, “ar”)) AND (LIMIT-TO (SUBJAREA, “AGRI”)) AND (LIMIT-TO (LANGUAGE, “English”))	5184
Filter application n.2	TITLE-ABS-KEY (mulberry *) OR TITLE-ABS-KEY (silkworm *) AND NOT TITLE-ABS-KEY (Antheraea)) AND (LIMIT-TO (DOCTYPE, “ar”)) AND (LIMIT-TO (SUBJAREA, “AGRI”)) AND (LIMIT-TO (LANGUAGE, “English”))	4598

^1^ Referred to years from 2000 to 2020.

**Table 2 insects-13-00568-t002:** Name of the considered clusters and words that compose them.

Cluster	Words
Cultivation	Crop, cultivar, fruit, germplasm, leaf, *Morus* sp. Pl, *M. alba*, *M. nigra*, mulberry, plant, root, seed, shoot, soil, species, variety, tree, water
Silkworm	Body, *Bombyx*, *B. mori*, breed, cocoon, diapause, diet, disease, DNA, egg, embryonic, expression, female, gene, gland, hemolymph, hormone, instar, larva, male, hybrid, immune, infection, instar, juvenile, larva, Lepidoptera, male, midgut, molting, moth, prothoracic, pupa, sex, shell, silkmoth, silkworm, spinning, stage, strain
Rearing	*B. bassiana*, feeding, heat, humidity, nutrition, *N. bombycis*, NucleoPolyhedroVirus, rearing, sericulture, stress, survival, temperature
Process	Drying, fermentation, industry, purification
Product	Acid, amino acid, anthocyanin, antioxidant, composition, diabetic, edible, enzyme, extract, fiber, fibroin, filament, flavonoid, food, juice, lipid, oil, peptide, phenol, polyphenol, powder, product, protein, quality, sericin, serine, silk, yield

**Table 3 insects-13-00568-t003:** Collaborations among countries.

Country	Total Number of Collaborations	Number of Partner Countries	Average Number of Collaborations per Country
Japan	142	37	3.9
USA	102	41	2.6
India	74	24	3.1
China	52	25	2.1
Italy	27	22	1.6

**Table 4 insects-13-00568-t004:** The most important words considering their slope, Pearson correlation coefficient and their relative increment in the past five years.

Word	Slope	Pearson Correlation Coefficient	Five Years Relative Changing [%]	Graph
Gene	18.2	0.85	+63%	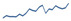
Protein	12.6	0.90	+68%	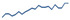
Acid	8.7	0.93	+172%	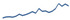
Fruit	7.3	0.92	+134%	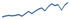
Extract	9.6	0.93	+159%	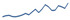
*Morus alba*	6.4	0.92	+118%	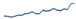
*Morus nigra*	3.3	0.91	+161%	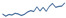
Antioxidant	8.6	0.97	+218%	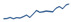
Silk	4.9	0.81	+47%	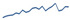
Expression	14.3	0.93	+106%	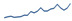
Anthocyanin	4.5	0.93	+270%	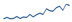
Phenolic	3.4	0.87	+278%	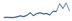
Polyphenol	1.9	0.91	+276%	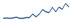
Flavonoid	3.0	0.90	+274%	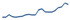
Immune	2.4	0.88	+307%	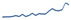
Juice	2.3	0.78	+234%	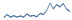

**Table 5 insects-13-00568-t005:** Most important words normalized per topic considering their slope, Pearson regression coefficient and their relative increment in the past five years.

Word	Slope	Pearson Regression Coefficient	Five Years Relative Changing [%]	Graph
Gene	2.30	0.54	+15%	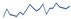
Protein	1.32	0.47	+16%	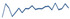
Acid	2.24	0.87	+101%	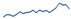
Fruit	1.99	0.85	+77%	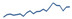
Extract	2.64	0.87	+96%	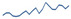
*Morus alba*	1.24	0.76	+50%	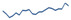
*Morus nigra*	0.83	0.76	+94%	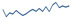
Antioxidant	2.73	0.96	+155%	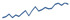
Silk	0.42	0.26	+3%	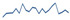
Expression	3.45	0.84	+53%	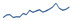
Anthocyanin	1.41	0.88	+194%	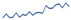
Phenol	1.50	0.85	+215%	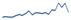
Polyphenol	0.64	0.89	+217%	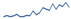
Flavonoid	0.94	0.88	+200%	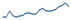
Immune	0.76	0.86	+212%	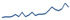
Juice	0.71	0.66	+166%	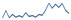

**Table 6 insects-13-00568-t006:** Clusters composition with the relative weight of each word and the overall relative weight of each cluster.

Cluster	Words	Cluster Relative Weight
Cultivation	Crop (1.6%), cultivar (1.4%), fruit (6.5%), germplasm (0.6%), leaf (15.2%), *Morus* sp.pl. (3.4%), *Morus alba* (6.5%), *Morus nigra* (2.9%), mulberry (32.2%), plant (7.7%), root (2.4%), seed (1.0%), shoot (1.3%), soil (2.0%), species (6.5%), tree (2.9%), variety (2.6%), water (3.3%)	23.6%
Silkworm	Body (1.8%), *Bombyx* (0.6%), *Bombyx mori* (10.7%), breed (1.6%), cocoon (3.4%), diapause (1.1%), diet (1.3%), disease (1.7%), DNA (1.9%), egg (2.3%), embryonic (0.5%), expression (7.4%), female (1.5%), gene (12.6%), gland (2.0%), hemolymph (1.1%), hormone (1.3%), hybrid (1.4%), immune (0.8%), infection (2.3%), instar (1.5%), juvenile (0.5%), larva (5.8%), Lepidoptera (1.5%), male (0.9%), metabolism (0.9%), midgut (1.2%), moth (0.8%), prothoracic (0.3%), pupa (2.7%), sex (0.8%), shell (0.9%), silkmoth (0.3%), silkworm (15.3%), spinning (2.9%) stage (2.4%), strain (2.9%), voltinism (1.0%)	46.4%
Rearing	*Beauveria bassiana* (5.1%), feeding (23.9%), heat (4.5%), humidity (2.7%), *Nosema bombycis* (5.1%), NucleoPolyhedroVirus (13.1%), rearing (8.0%), sericulture (7.4%), stress (9.5%), survival (7.2%), temperature (13.5%)	6.1%
Process	Drying (47.6%), fermentation (17.2%), industry (14.0%), purification (12.1%)	1.6%
Product	Acid (8.6%), aminoacid (2.4%) anthocyanin (3.4%), antioxidant (6.9%), composition (7.7%), diabetic (2.5%), edible (0.8%), enzyme (2.9%), extract (8.7%), lipid (4.3%), fiber (0.7%), fibroin (1.5%), filament (0.8%), flavonoid (2.3%), food (3.3%), juice (1.9%), oil (1.4%), peptides (1.9%), phenol (3.5%), polyphenol (1.4%), powder (1.4%), product (2.1%), protein (18.3%), quality (3.2%), sericin (1.5%), serine (1.1%), silk (8.2%), yield (3.5%),	22.2%

**Table 7 insects-13-00568-t007:** Analysis of cluster considering their slope, Pearson regression coefficient, and their relative increment in the past five years.

Cluster	Slope	Pearson Regression Coefficient	Five Years Relative Changing [%]	Graph
Cultivation	83.4	0.93	+80%	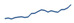
Process	7.4	0.89	+144%	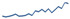
Product	90.0	0.98	+117%	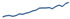
Rearing	18.9	0.87	+71%	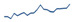
Silkworm	105.7	0.79	+33%	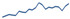

**Table 8 insects-13-00568-t008:** Analysis of clusters normalized per cluster by considering their slope, Pearson regression coefficient, and their relative increment in the past five years.

Cluster	Slope	Pearson Regression Coefficient	Five Years Relative Changing [%]	Graph
Cultivation	11.57	0.79	+25%	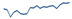
Process	1.64	0.82	+73%	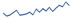
Product	18.53	0.93	+55%	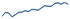
Rearing	1.55	0.40	+16%	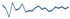
Silkworm	−6.67	−0.35	−10%	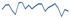

**Table 9 insects-13-00568-t009:** Calculated impact factor of the most frequent words for the five clusters. The words are sorted in descending order considering their weights according to Equation (1); h-indices in brackets.

Cluster	Words	Cluster Mean
Cultivation	Mulberry (89), leaf (77), plant (94), *Morus alba* (75), species (84)	84
Silkworm	Silkworm (78), gene (95), *Bombyx mori* (77), expression (73), larva (63)	77
Rearing	Feeding (62), temperature (55), nucleopolyhedroviral (35), stress (60), rearing (61)	51
Process	Drying (38), fermentation (39), industry (42), purified (66)	46
Product	Protein (82), extract (82), acid (97), silk (56), antioxidant (77)	79

**Table 10 insects-13-00568-t010:** Number of interrelationships among clusters.

	Cultivation	Process	Product	Rearing	Silkworm
Cultivation	15,203	-	-	-	-
Process	17,010	25	-	-	-
Product	19,756	2156	13,028	-	-
Rearing	5415	520	6186	924	-
Silkworm	20,626	2396	50,422	12,135	61,864

## Data Availability

Data presented in this study are available on request from the corresponding author. The data are not publicly available because they can be used by the Operational Group “Serinnovation” for future economic applications.

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
