# Peer review of "Bibliometric Analysis of Trends in Mulberry and Silkworm Research on the Production of Silk and Its By-Products"

_insects, 2022, doi:10.3390/insects13070568_

Round 1

Reviewer 1 Report

The manuscript "Bibliometric Analysis of Trends in Mulberry and Silkworm Research for Silk Production", written by Domenico Giora, Giuditta Marchetti, Silvia Cappellozza, Alberto Assirelli, Alessio Saviane, Luigi Sartori and Francesco Marinello, presents a comprehensive study based on a text-mining analysis of the 421 words in the title, abstract and keywords of articles on silk production over the two last decades. The work described in the manuscript fits perfectly within the framework of the special issue “Silkworm and Silk: Traditional and Innovative Applications” of the journal Insects. The bibliometric analysis is interesting and important as it highlights knowledge gaps in silk production, especially the paucity of research papers on the topic of automation of processes.

The methodology of the biometric analysis together with the statistical analysis is explained in detail and supported by a schematic of the whole analysis (Figure 1). The authors explained the elaboration of the article (tokenisation, identification of the most frequently recurring words in the last two decades), the biometric analysis was evaluated and represented by a word-word connection matrix based on the list of 100 most frequently recurring words. To study the inter-connection of nouns, the authors performed a cluster analysis.

Since the authors emphasise that sericulture cultivation has transcended its usual boundaries and is linked to other disciplines, e.g. medicine, cosmetics, engineering, fruit processing, and since the authors have also considered the clusters of 'product' and 'cultivation' with important words related to leaf and fruit constituents, I would suggest changing the title to “Bibliometric Analysis of Trends in Mulberry and Silkworm Research on the Production of Silk and By-Products”.

My suggestion is also supported by the authors' findings that the noun 'leaf' shows good growth but is not so constant in absolute numbers over time. In contrast, researchers' interest in mulberry fruit has increased in both absolute and relative numbers.

Overall, the study is well designed and conducted; I congratulate the author for undertaking such a challenging biometric analysis which provided quantitative indicators for researchers, entrepreneurs, and scholars. In my opinion, the manuscript is acceptable for publication.
Some textual improvements are listed below:

Page 2, Line 53: Rewrite B. mori to Bombyx mori as it is named the first time in Introduction section.

Page 3, line 97: Anthocyanins are only one group of flavonoids (along with flavonols and flavan-3-ols). Out of flavonols the predominant are quercetin and kaepferol glycosides. Among other phenolics hydroxycinnamic acids derivatives are presented in high amount with the predominant chlorogenic acid.

Revise the sentence: »…contain relatively high amounts of flavonoids [40–42], phenols [42–44] and anthocyanins« to »contain relatively high amounts of phenols [42–44] in particular flavonoids [40–42] with predominant quercetin and kaempferol glycosides as well as anthocyanins.

Page 3, line 114: revise Sciences to sciences.

Page 16, line 541: Dot is missing.

Author Response

Response to Reviewer 1 Comments

  • Since the authors emphasise that sericulture cultivation has transcended its usual boundaries and is linked to other disciplines, e.g. medicine, cosmetics, engineering, fruit processing, and since the authors have also considered the clusters of 'product' and 'cultivation' with important words related to leaf and fruit constituents, I would suggest changing the title to “Bibliometric Analysis of Trends in Mulberry and Silkworm Research on the Production of Silk and By-Products”.

The title was changed according to the referee’s suggestion.

  • Some textual improvements are listed below:

Page 2, Line 53: Rewrite B. mori to Bombyx mori as it is named the first time in Introduction section.

Bombyx mori was already written in full at line 42

 Page 3, line 97: Anthocyanins are only one group of flavonoids (along with flavonols and flavan-3-ols). Out of flavonols the predominant are quercetin and kaepferol glycosides. Among other phenolics hydroxycinnamic acids derivatives are presented in high amount with the predominant chlorogenic acid. Revise the sentence: »…contain relatively high amounts of flavonoids [40–42], phenols [42–44] and anthocyanins« to »contain relatively high amounts of phenols [42–44] in particular flavonoids [40–42] with predominant quercetin and kaempferol glycosides as well as anthocyanins.

The sentence was changed accordingly.

Page 3, line 114: revise Sciences to sciences.

It was done.

Page 16, line 541: Dot is missing.

Thank you; it was added.

Reviewer 2 Report

This paper makes a statistical analysis of sericulture and related literature published in the past 20 years. The results provide sericulture researchers with a comprehensive understanding of the development trend and research hot spots of sericulture research. I personally think this research is very meaningful and worth publishing. Here, could the author make a statistical analysis on the level of literature publication? According to my personal feeling, the overall level of sericulture publications is not very high, and there are few articles published in CNS. Through the publication analysis of high-level articles, we can see which filed of sericulture are at a relatively high level in the field of Entomology. In the discussion section, many descriptions are more like the contents of the results section. Whether to compare and analyze with similar analysis articles of other species.

Author Response

  • This paper makes a statistical analysis of sericulture and related literature published in the past 20 years. The results provide sericulture researchers with a comprehensive understanding of the development trend and research hot spots of sericulture research. I personally think this research is very meaningful and worth publishing. Here, could the author make a statistical analysis on the level of literature publication? According to my personal feeling, the overall level of sericulture publications is not very high, and there are few articles published in CNS. Through the publication analysis of high-level articles, we can see which filed of sericulture are at a relatively high level in the field of Entomology. 

We appreciate the reviewer's proposal to evaluate the level of publications in the literature.

For us it was not easy to define a proper metrics to evaluate the level of literature publication. We eventually decided to implement the h-index, which is widely known and used for such kind of comparisons. Thus, we have added the following paragraphs:

Page 6, lines from 243 to 248

“The authors analyzed the impact of the five most frequent terms of each cluster, according to Equation 1. To this end, the impact of the papers related to such terms was quantified in terms of Hirsch’s h-index. Thus, for a given term, the h-index has been here calculated by counting the number h of publications including that term and cited at least that same number h of times. The average was then computed for the five terms of each cluster.”

Page 13, lines from 406 to 418

The objective of this analysis was to derive sound information about the impact that the most frequent terms generated on scientific research. The proposed impact factors of considered word are reported in Table 9.

As reported in Table 9, the “Cultivation” and “Product” clusters are the ones ex-hibiting the highest impact, highlighting a vivid interest by the scientific community on these topics. Also the “Silkworm” cluster gives evidences of a significant impact: in this case, the high h-index is to be related also to the highest number of papers published on this topic since 2000. A lower interest is apparently arising from researches related to the process or to entomological aspects. With respect to the specific terms, “acid” (97) and “gene” (95) gained the highest impact, having collected more than 35000 citing documents each.”

  • In the discussion section, many descriptions are more like the contents of the results section.

Please see the changes in the text of the Results and Discussion sections in the newly submitted paper.

  • Whether to compare and analyze with similar analysis articles of other species.

The proposal to compare the results of our research with publications from other species is very interesting. However, we believe that this could enlarge the analysis of this paper too much and bring us out of the scope. A comparison among the research on the silkworm and on other insects or animals can be the basis for another future publication

Reviewer 3 Report

The publication is prepared very carefully, precisely describing the issues related to silkworms and silk. Apart from the introduction that is too long and should be shortened, I have nothing more to comment on. The work follows the scheme that is known from other publications, e.g. doi:10.3390/EN13143714

Author Response

Response to Reviewer 3 Comments

  • The publication is prepared very carefully, precisely describing the issues related to silkworms and silk. Apart from the introduction that is too long and should be shortened, I have nothing more to comment on. The work follows the scheme that is known from other publications, e.g. doi:10.3390/EN13143714

We appreciate the reviewer's suggestion to shorten the Introduction. Please see the changes in the text of the Introduction in the enclosed paper.